# Text2Smell: Emergent Representations of Human Olfactory Perception in Large Language Models

## Abstract

Large language models (LLMs) trained exclusively on text have recently demonstrated emergent capacities to predict human judgments across several sensory-related tasks, suggesting that linguistic co-occurrence statistics implicitly encode aspects of human sensory experience. Yet, whether such models capture the structure of olfactory perception, one of the most complex and least understood human senses, remains unknown. In this work, we investigate whether state-of-the-art LLMs can predict human smell perception purely from linguistic cues and how their representations compare to those of molecular transformer models explicitly trained on chemical structure. We prompt LLMs to provide perceptual olfactory ratings to odorants, and evaluate their outputs against human ratings across several datasets. Surprisingly, we find that LLMs exhibit strong alignment with human perceptual judgments, comparable to, and in most cases exceeding, the performance of specialized molecular transformers. These results indicate that linguistic knowledge alone carries rich latent structure about human olfaction, bridging the gap between language and chemical perception. Our findings position LLMs as powerful linguistically grounded perceptual models and open new directions for studying sensory grounding and cross-modal representation learning through language.

## 1 Introduction

Recent advances in large language models (LLMs) have shown that models trained purely on text can acquire surprisingly broad sensory knowledge, from visual attributes to sound semantics (Marjieh et al., 2024; Zhang et al., 2022a; Siedenburg & Saitis, 2023). Through large-scale language pretraining, these models develop internal representations that capture rich semantic, perceptual, and commonsense regularities present in human descriptions of the world (Shiono et al., 2025). Although they are never exposed to raw sensory data, LLMs demonstrate an emergent capacity to reason about sensory concepts and make judgments that are consistent with human perception, for example, describing textures as "rough" or "smooth" (Tu et al., 2025), or predicting emotional valence from speech (Lalk et al., 2025). This suggests that the linguistic co-occurrence patterns embedded in massive text corpora implicitly encode perceptual relationships, providing a bridge between symbolic knowledge and the structure of human experience. Understanding the extent and limits of this implicit sensory grounding is therefore crucial for advancing language models toward genuine perceptual understanding and for uncovering how human sensory knowledge is reflected in language itself.

Among all human senses, *olfaction*, the sense of smell, represents a uniquely complex and challenging domain for computational modeling. Unlike vision or audition, where perceptual responses are grounded in well-characterized physical dimensions such as wavelength or frequency, the mapping between molecular structure and olfactory experience remains poorly understood (Saini & Ramanathan, 2022). Recent works in chemoinformatics, computational olfaction, and molecular transformer models have aimed to predict odor qualities from molecular features using handcrafted descriptors, graph-based neural networks, or large transformer-based encoders (Lee et al., 2023; Ravia et al., 2020; Taleb et al., 2024; Lötsch et al., 2019). While these approaches capture chemical commonalities, they often fail to fully reflect the subjective nature of human smell perception. For instance, two molecules with similar structural features may evoke vastly different perceptual impressions, and conversely, perceptually similar odors may arise from structurally dissimilar

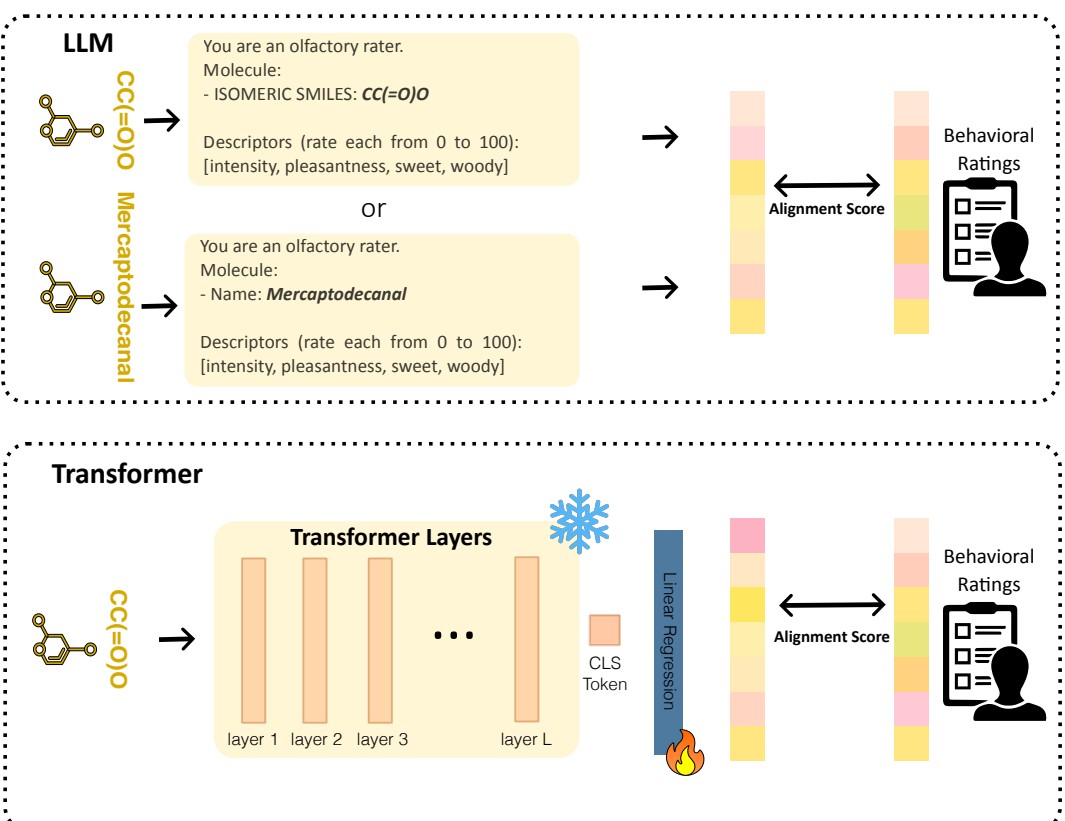

Figure 1: **Overview of the main methodology.** Top: LLMs are prompted with either the common names or SMILES representations of molecules and instructed to provide perceptual ratings similar to human judgments. Bottom: Representations of the same molecules are extracted from specialized molecular transformers trained on chemical structures.

compounds (Sharma et al., 2019; Sell, 2006; Boesveldt et al., 2010). This long-standing "structure–odor gap" motivates an alternative perspective: rather than predicting perception solely from molecular structure, *can we leverage the rich perceptual knowledge embedded in language to predict how humans experience the smells of different odorants with diverse molecular structures?* In this work, we explore whether LLMs can predict human olfactory perception purely from linguistic cues, and how their representations compare to those of specialized chemical models. We prompt state-of-the-art LLMs (e.g., GPT, Gemini) to rate odorants along perceptual dimensions and semantic descriptors (e.g., pleasantness, sweet, woody, fishy), given either the odorant's common name or molecular representations. We then evaluate their outputs against human behavioral ratings from large-scale psychophysical datasets and against predictions derived from pre-trained molecular transformers. Our analyses reveal that LLMs exhibit robust alignment with human olfactory judgments, often outperforming the predictive performance of chemical structure–based models.

In this paper, we make three main contributions:

- We introduce a systematic framework for evaluating the alignment between LLMs and human olfactory perception, combining large-scale psychophysical data with model-prompted odor ratings

- We provide the first direct comparison between LLMs and molecular transformer models in predicting perceptual ratings across multiple semantic dimensions.

- We demonstrate that LLMs, despite never being explicitly fine-tuned for this task, can serve as linguistically grounded perceptual models, capturing cross-modal regularities that emerge from human descriptions of sensory experience.

| Dataset | # of Participants | # of Odorants | Rating Type | Rating Range | # of Descriptors |
|---|---|---|---|---|---|
| Keller (Keller & Vosshall, 2016) | 55 | 420 | Continuous | [0,100] | 23 |
| Sagar (Sagar et al., 2023) | 3 | 125 | Continuous | [-1,1] | 15 |
| Leffingwell (Sanchez-Lengeling et al., 2019) | 1 | 3522 | Binary | {0,1} | 113 |
| Snitz (Snitz et al., 2013) | 139 | 83 | Similarity | [0,1] | NA |

Table 1: **All the datasets used in this study.** The number of odorants and perceptual descriptors shown corresponds to those retained for analysis following data cleaning and curation.

Together, these contributions bridge the gap between language and olfaction, highlighting that perceptual knowledge is implicitly encoded in linguistic co-occurrence patterns. Our work opens new directions for multimodal research on sensory grounding in language, suggesting that LLMs can function not only as generators of linguistic structures but also as proxies for human perceptual cognition.

## 2 Related Works

This study investigates how well LLMs align with human perception in the domain of olfaction. We situate our work at the intersection of three research threads: (i) using LLMs to model human perception and cognition, (ii) employing LLMs for molecular representation and reasoning, and (iii) modeling olfactory perception through both linguistic and chemical representations. After describing each of these three threads, we describe the research gap that we will address in this paper.

**LLMs as Models of Human Perception.** A growing body of research examines whether LLMs capture aspects of human perception and subjective experiences. Marjieh et al. (2024) show that LLMs can predict human similarity judgments across six sensory modalities, including color, pitch, timbre, loudness, taste, and speech sounds, revealing that language-only models recover canonical perceptual structures even without direct sensory grounding. However, Xu et al. (2025) systematically analyzed conceptual representations from non-sensorimotor dimensions (e.g., valence, dominance) to sensorimotor domains (e.g., auditory, visual, olfactory) across a broad range of concepts, and observed a gradual decline in similarity between LLM-derived and human-derived representations, with the strongest divergences emerging in sensorimotor domains. Similarly, Hicke et al. (2025) analyzes sensory language in model-generated narratives and reports systematic mismatches with human usage; all models generate stories that differ significantly from human usage of sensory language, but the direction of these differences varies considerably between model families.

**LLMs for Molecular Representation.** Recent efforts have extended LLMs beyond natural language to the chemical and molecular domains, enabling their application to chemistry-related tasks such as reaction prediction (Liu et al., 2024; M. Bran et al., 2024). In more recent works, Zhuang et al. (2025) introduces a framework for molecular structure elucidation, extending LLM's coverage of the chemical structure space. In property prediction, Xian et al. (2025) presents MolRAG, a retrieval-augmented generation framework that grounds textual reasoning in molecular databases, achieving stronger generalization and interpretability. Extending to multi-objective optimization, Dey et al. (2025) introduce the first instruction-tuned family of LLMs for molecule optimization, which outperforms state-of-the-art baselines on both in- and out-of-domain tasks.

**Modeling Olfactory Perception with LLMs** Olfactory perception remains largely underexplored in machine learning and, in particular, in the study of LLMs. To the best of our knowledge, the only prior work examining LLM–human alignment in this domain is SniffAI (Zhong et al., 2024). However, SniffAI focuses solely on textual odor descriptions, with no link to the underlying chemical or molecular structures, and is based on a small dataset of 20 odor objects from 4 families rated by 40 participants. In contrast, our work evaluates alignment using the actual molecular stimuli that constitute odorants, providing a more direct examination of how LLM representations relate to human olfactory perception.

| Model | Input Type | # of Parameters | Base Model | Dataset | Layers |
|-------|-----------|-----------------|-----------|---------|--------|
| MoLFormer-XL (Ross et al., 2022) | SMILES | 45M | Transformer | ZINC, PubChem | 12 |
| MTL-BERT (Zhang et al., 2022b) | SMILES | 6M | BERT | ChEMBL | 8 |
| ChemBERTa (Chithrananda et al., 2020) | SMILES | 44M | RoBERTa | PubChem | 6 |

Table 2: **The pre-trained molecular transformer models used in this study and their characteristics**. All models are publicly available via HuggingFace. The "Dataset" column indicates the training dataset used for each model.

**Research Gap**  Although understanding sensory encoding in LLMs is important, current research is still limited. Studies on LLMs and human perception have focused primarily on visual, auditory, or tactile modalities, without examining the uniquely complex case of olfaction. In parallel, molecular transformer models achieve strong performance in structural and physicochemical prediction tasks but lack access to the subjective and linguistic dimensions of human smell perception (Taleb et al., 2024). Previous attempts to connect language and olfaction, such as SniffAI (Zhong et al., 2024), rely solely on textual odor descriptions and small-scale datasets, without linking model predictions to actual molecular stimuli or human psychophysical data. Our work bridges these gaps by directly comparing LLM-derived and molecule-based representations of olfactory perception, providing a large-scale evaluation of linguistic sensory alignment in the olfactory domain.

## 3 Background

### 3.1 Olfactory Perception Task

Olfactory perception refers to the subjective experience of smell and is typically examined through psychophysical experiments. Existing datasets have been collected using three main approaches. In the first, participants smell individual odorants and provide continuous ratings along perceptual dimensions (e.g., pleasantness, intensity, sweetness, and woodiness). In the second approach, participants provide binary ratings for a predefined set of descriptors, indicating whether a perceptual quality (e.g., fruity, sweet) is present (1) or absent (0) for each odorant. In the third approach, participants, given a range, judge the similarity between pairs of odorants, assigning numerical scores that quantify how similar two odors smell.

Throughout this paper, we refer to these responses collectively as perceptual or behavioral ratings. Together, these ratings characterize how humans perceive molecular stimuli and form the foundation for modeling perceptual similarity and structure in the olfactory domain.

### 3.2 Molecular Representations

Each odorant can consist of a single molecule, referred to as a mono-molecular odorant, or a combination of multiple molecules, forming a mixture. In this work, we focus exclusively on odorants composed of a single molecule. Each such odorant can be represented either by its common name (e.g., phenylethyl alcohol) or by string-based molecular encodings such as SMILES (Simplified Molecular Input Line Entry System) (Weininger, 1988) . SMILES encodes a molecule's structure as a linear string describing its atoms and bonds. Chemical models typically rely on these structural representations to infer molecular properties, whereas LLMs can process both textual names and symbolic strings. This dual encoding enables us to investigate how linguistic and structural information each contributes to predicting human olfactory perception.

### 3.3 Datasets

We use the publicly available datasets summarized in Table 1. These datasets contain detailed information about odorants and their constituent molecules, along with corresponding behavioral ratings from human participants. Each dataset was collected using one of the psychophysical approaches described in Section 3.1. For all analyses, we use the average response across repetitions and participants for each odorant.

For the `Keller` (Keller & Vosshall, 2016), `Sagar` (Sagar et al., 2023), and `Leffingwell` (Sanchez-Lengeling et al., 2019) datasets, this yields a behavioral rating matrix $\mathbf{R}^{\text{behavioral}} \in \mathbb{R}^{N \times P}$, where $N$ denotes the number of odorants and $P$ the number of perceptual descriptors in that dataset. For `Keller` dataset, we included only the odorants that were rated by all participants and when multiple concentrations were available, we selected the behavioral ratings corresponding to the concentration of 0.001%. Missing (`NaN`) values were replaced with zeros, and the final behavioral rating for each odorant was obtained by averaging across all participants. For the `Sagar` dataset, we included only odorants that consisted of single molecules (mono-molecular stimuli). Additionally, we retained only the odorants and descriptors that were common across all participants. For `leffingwell` dataset, we used it as provided in (Sanchez-Lengeling et al., 2019), without additional curation or filtering. All odorant and descriptor associations were retained in their original binary form.

For the `Snitz` (Snitz et al., 2013) dataset, which are based on pairwise similarity judgments, we construct a sparse similarity matrix $\mathbf{S}^{\text{behavioral}} \in \mathbb{R}^{N \times N}$ where each entry $S_{ij}$ represents the perceived similarity between odorants $i$ and $j$ for the pairs that were evaluated by participants. For this dataset, we included only pairs of odorants in which both stimuli were mono-molecular. Not all odorant pairs have associated similarity ratings, resulting in partially observed matrices. All datasets used in this study are publicly available and can be accessed either through their corresponding GitHub repositories or via the Pyrfume library (Hamel et al., 2024).

## 4 Methodolgy

Our objective is to evaluate the alignment between human and LLMs' olfactory responses when perceiving odorants, and to compare this alignment with representations derived from pre-trained molecular models of chemical structure. In this section, we introduce the models used in this study and how we measure the alignment according to the data provided.

All the datasets used in this study are publicly available through their corresponding GitHub repositories or via the Pyrfume library (Hamel et al., 2024). We access the proprietary models via their APIs, and the open-source models through HuggingFace. All other models used in our work are linear, which makes both their training and inference computationally lightweight. The code and instructions to reproduce all experiments and results are available at `https://anonymous.4open.science/r/Mol2Smell-FE70/`.

### 4.1 Models

**LLMs**  We evaluate seven state-of-the-art LLMs, spanning both open-source and proprietary models, by prompting them to generate perceptual ratings corresponding to those provided by humans. The proprietary models includes Gemini-2.5-flash[1], GPT-5-mini[2], and Claude-sonnet-4-5[3]. The open-source models includes Qwen-3.5-27B[4], Gemma-4-31B-it[5], OLMo-3-1125-32B[6], and Llama-3.3-70B-Instruct[7].

The open-source models were selected to cover recent model families with different designs and training philosophies. Qwen and Gemma represent highly optimized recent model families that are commonly positioned as having strong instruction-following, in-context learning, and reasoning-oriented ca-

---

[1]`https://deepmind.google/models/gemini/flash/`
[2]`https://platform.openai.com/docs/models/gpt-5-mini`
[3]`https://www.anthropic.com/news/claude-sonnet-4-5`
[4]`https://huggingface.co/Qwen/Qwen3.5-27B`
[5]`https://huggingface.co/google/gemma-4-31B-it`
[6]`https://huggingface.co/allenai/Olmo-3-1125-32B`
[7]`https://huggingface.co/meta-llama/Llama-3.3-70B-Instruct`

pabilities (Google, 2026a;b; Yang et al., 2025; Qwen Team, 2026). These capabilities may be attributed to a combination of modern architectural design choices, context utilization mechanisms, and post-training strategies. In contrast, OLMo and Llama provide complementary dense decoder-only transformer baselines with different model scales, training recipes, and post-training strategies. This selection allows us to examine whether models such as Gemma and Qwen, which are commonly associated with stronger general-purpose reasoning and instruction-following capabilities, produce perceptual rating patterns that are more aligned with human judgments.

Depending on the dataset type (absolute ratings or similarity comparisons, see Section 3.3), we designed two types of prompt templates, which were filled with the molecular representations, the target rating range, and the relevant perceptual descriptors (e.g., intensity, pleasantness, fruity). Descriptor sets and rating ranges are derived from the original human datasets, with a single exception: in the `Leffingwell` dataset the human labels are binary, but we prompt the LLMs to return probabilities in the range [0,1] for each descriptor. Our motivation for this choice was to allow greater flexibility in designing evaluation metrics that are more meaningful, especially since converting probabilities to binary values is straightforward, whereas the reverse is not. For each dataset, 3 independent repetitions were generated by each LLM for every human rating, and the results were aggregated by computing their mean. These outputs were then represented as numerical matrices, where for each LLM $l \in L$, we obtain a matrix $\mathbf{R}_l^{\text{LLM}} \in \mathbb{R}^{N \times P_d}$, with $N$ denoting the number of odorants and $P_d$ the number of perceptual descriptors for dataset $d$. This matrix subsequently serves as the basis for downstream aggregation and alignment analyses with human perceptual ratings. Regarding the temperature parameter, when applicable, we followed the recommended best-practice settings provided for each model.

**Transformers** We use three state-of-the-art encoder-only transformer models (as summarized in Table 2) pre-trained on large-scale chemical datasets, to extract computational representations of odorants. These models, publicly available via HuggingFace[8], are trained using self-supervision and have demonstrated strong performance on tasks such as property and reaction prediction.

## 4.2 Generating Model Responses

Model responses can be obtained through two primary approaches: (i) prompting LLMs to generate textual responses, and (ii) extracting internal embeddings by providing the models with structured prompts or molecular inputs.

For LLMs, we obtained model outputs directly as textual responses generated from the prompts described above. For transformer-based molecular models, we provided each model with string-based representations of individual molecules in the form of SMILES. From the resulting hidden states, we extracted the classification token (`[CLS]`) from the final transformer layer as the molecular representation. For each model $m \in M$, this procedure yields a representation matrix $\mathbf{R}_m^{\text{transformer}} \in \mathbb{R}^{N \times D_m}$, where $N$ denotes the number of odorants and $D_m$ is the embedding dimensionality of model $m$.

It is also worth noting that these two model classes require different evaluation approaches: molecular transformers are typically encoder-only and therefore cannot generate direct textual responses, whereas proprietary LLMs generally do not provide access to internal embeddings. As a result, we use embedding-based evaluation for molecular transformers and generation-based evaluation for LLMs.

## 4.3 Evaluation

**Alignment Evaluation for LLMs** Since the LLMs were prompted to produce numerical ratings, their alignment with human perceptual responses can be directly assessed. For datasets containing continuous ratings or similarity judgments between pairs of odorants, we measure alignment using the *Pearson correlation coefficient (r)* between LLM-derived and human-derived ratings. Otherwise, for the `Lefingwell` dataset, where human responses are provided as binary labels for predefined descriptors, we assess alignment using the *AUC-ROC* score.

---

[8]`https://huggingface.co/models`

| Model family | Model | Keller | Sagar | Snitz | Leffingwell |
|---|---|---|---|---|---|
| | | $r$ | $r$ | $r$ | ROC-AUC |
| Transformers | MoLFormer | $0.19 \pm 0.03$ | $0.21 \pm 0.04$ | $0.33 \pm 0.001$ | $0.84 \pm 0.01$ |
| | MTL-BERT | $0.20 \pm 0.03$ | $0.20 \pm 0.04$ | $0.60 \pm 1.66$ | $0.84 \pm 0.01$ |
| | ChemBERTa | $0.16 \pm 0.03$ | $0.18 \pm 0.04$ | $0.11 \pm 0.31$ | $0.79 \pm 0.01$ |
| LLMs | Gemini-2.5-flash | $0.39 \pm 0.04$ | $0.32 \pm 0.04$ | $0.76 \pm 9.91$ | $0.76 \pm 0.01$ |
| | GPT-5-mini | $0.35 \pm 0.04$ | $0.27 \pm 0.05$ | $0.72 \pm 1.74$ | $0.75 \pm 0.01$ |
| | Claude-sonnet-4-5 | $0.35 \pm 0.03$ | $0.25 \pm 0.03$ | $0.69 \pm 0.00$ | $0.79 \pm 0.01$ |
| | Gemma-4-31B-it | $0.26 \pm 0.04$ | $0.25 \pm 0.05$ | $0.76 \pm 0.00$ | $0.64 \pm 0.01$ |
| | Qwen-3.5-27B | $0.17 \pm 0.04$ | $0.21 \pm 0.05$ | $0.64 \pm 0.00$ | $0.61 \pm 0.01$ |
| | Llama-3.3-70B-Instruct | $0.07 \pm 0.02$ | $0.08 \pm 0.03$ | $0.75 \pm 0.00$ | $0.56 \pm 0.01$ |
| | OLMo-3-1125-32B | $0.04 \pm 0.02$ | $0.06 \pm 0.02$ | $0.52 \pm 0.00$ | $0.51 \pm 0.01$ |

Table 3: **Alignment scores across models and datasets.** LLMs outperform in similarity-based tasks and in predicting continuous ratings over a limited set of descriptors. In contrast, specialized molecular transformers achieve higher performance when richer chemical information is available. For details of the alignment score, see Section 4.3. Higher values indicate better alignment.

**Alignment Evaluation for Transformers**  Since it is not possible to directly measure alignment between embeddings and human ratings, linear models (Yamins & DiCarlo, 2016) are widely employed in the machine learning community (d'Ascoli et al., 2025; Zhang et al., 2025; Toneva & Wehbe, 2019) to assess this alignment. These models minimize the influence of additional model complexity, enabling a more direct and interpretable evaluation of representational correspondence. Following this standard approach, when the absolute continuous or binary ratings for odorants are provided, we first train linear models to predict the continuous or binary multi-target ratings assigned to odorants. We then use the trained models to predict held-out test targets, aggregate the predictions across the test sets of all 10 folds. Finally, we compute the *Pearson correlation coefficient (r)* for datasets containing continuous ratings and the *AUC-ROC* score for datasets with binary ratings, comparing the targets predicted from embeddings with human ratings, matching the same evaluation procedure used for the LLMs. For the similarity datasets, instead of training linear models, we compute cosine similarity between pairs of extracted representations and then evaluate alignment by correlating these similarities with those provided by humans to obtain the final *Pearson correlation coefficient (r)*.

**Unified Alignment Score.**  Throughout this paper, we use the term **alignment score** to denote both the AUC-ROC and Pearson correlation coefficient metrics, as each quantifies the correspondence between model predictions and human responses, tailored to the respective output type. When interpreting the results, it should be noted that the chance levels of the two metrics are not the same. AUC-ROC has an expected chance value of 0.5, whereas Pearson correlation has a chance value of 0.

## 5  Results

In this section, we explore the alignment between LLMs and humans through three research questions:
**RQ1:** How well do LLM-generated perceptual ratings align with human ratings, compared to molecular transformers across multiple datasets and output representation types?
**RQ2:** Does the type of input representation, chemical structure (SMILES) versus common molecular name, affect the perceptual outputs of LLMs?
**RQ3:** Where do the observed alignment differences originate within LLMs across different input types?

**Alignment between LLMs and Humans.**  Table 3 summarizes the alignment results across all datasets and models. As shown, the proprietary LLMs, including Gemini-2.5-flash, GPT-5-mini, and Claude-sonnet-4-5, outperform the molecular transformer baselines on the three datasets with continuous or similarity-based

| Model | Keller | | Sagar | | Snitz | | Leffingwell | |
|---|---|---|---|---|---|---|---|---|
| | SMILES | Name | SMILES | Name | SMILES | Name | SMILES | Name |
| Gemini-2.5-flash | $0.39 \pm 0.04$ | $0.39 \pm 0.03$ | $0.32 \pm 0.04$ | $0.33 \pm 0.03$ | $0.76 \pm 9.91$ | $0.73 \pm 2.89$ | $0.76 \pm 0.01$ | $0.76 \pm 0.01$ |
| GPT-5-mini | $0.35 \pm 0.04$ | $0.38 \pm 0.04$ | $0.27 \pm 0.05$ | $0.32 \pm 0.04$ | $0.71 \pm 1.74$ | $0.71 \pm 3.99$ | $0.75 \pm 0.01$ | $0.77 \pm 0.01$ |
| Claude-sonnet-4-5 | $0.35 \pm 0.03$ | $0.41 \pm 0.03$ | $0.25 \pm 0.03$ | $0.31 \pm 0.03$ | $0.69 \pm 0.00$ | $0.67 \pm 0.00$ | $0.79 \pm 0.01$ | $0.79 \pm 0.01$ |
| Gemma-4-31B-it | $0.26 \pm 0.04$ | $0.40 \pm 0.03$ | $0.25 \pm 0.05$ | $0.33 \pm 0.04$ | $0.76 \pm 0.00$ | $0.62 \pm 0.00$ | $0.64 \pm 0.01$ | $0.72 \pm 0.01$ |
| Qwen-3.5-27B | $0.17 \pm 0.04$ | $0.38 \pm 0.04$ | $0.21 \pm 0.05$ | $0.26 \pm 0.05$ | $0.64 \pm 0.00$ | $0.58 \pm 0.00$ | $0.61 \pm 0.01$ | $0.71 \pm 0.01$ |
| Llama-3.3-70B-Instruct | $0.07 \pm 0.02$ | $0.28 \pm 0.04$ | $0.08 \pm 0.03$ | $0.31 \pm 0.04$ | $0.75 \pm 0.00$ | $0.68 \pm 0.00$ | $0.56 \pm 0.01$ | $0.63 \pm 0.01$ |
| OLMo-3-1125-32B | $0.04 \pm 0.02$ | $0.13 \pm 0.02$ | $0.06 \pm 0.02$ | $0.15 \pm 0.02$ | $0.52 \pm 0.00$ | $0.47 \pm 0.00$ | $0.51 \pm 0.00$ | $0.56 \pm 0.01$ |

Table 4: **Alignment scores across LLMs and datasets for two different input types.** Rows show models, while columns show dataset-specific performance under SMILES-based and name-based inputs. For Keller, Sagar, and Snitz, the metric is Pearson correlation ($r$); for Leffingwell, the metric is ROC-AUC.

perceptual ratings, namely `Sagar`, `Keller`, and `Snitz`, while on `Leffingwell`, where binary perceptual labels are provided, their performance is comparable to that of the molecular transformers. This result is interesting given that LLMs have never been fine-tuned for olfactory perception prediction, yet their inferred perceptual ratings show a substantially higher correspondence with human olfactory judgments.

Among the open-source models, Gemma-4-31B-it and Qwen-3.5-27B remain the strongest models across the three continuous/similarity-rating datasets, `Sagar`, `Keller`, and `Snitz`, either outperforming the molecular transformer baselines or achieving comparable performance. In contrast, Llama-3.3-70B-Instruct and OLMo-3-1125-32B perform above random but below the molecular transformer baselines. Importantly, this ordering does not follow parameter count: Llama-3.3-70B is larger than both Gemma-4-31B-it and Qwen-3.5-27B, yet performs worse on this task. This suggests that odor-prediction performance is not driven by model scale alone. Instead, the observed differences may reflect a combination of factors, including model-family design, instruction-following behavior, in-context learning ability, and post-training strategies. In particular, Gemma and Qwen are commonly positioned as recent model families with strong general-purpose reasoning and instruction-following capabilities, which may help them better interpret molecular representations and map them to perceptual descriptors. The similar behavior of Llama and OLMo, both dense decoder-only transformer models, suggests that architectural family and training recipe may also influence how well models transfer general language knowledge to olfactory prediction.

Overall, model performance appears to depend not only on model architecture, but also on the structure of the evaluation dataset. For the similarity-based task in `Snitz`, LLMs generally outperform molecular transformers, suggesting that they may better capture global or relational perceptual judgments. In contrast, for the binary descriptor prediction task in `Leffingwell`, molecular transformers perform slightly better than LLMs. This difference may partly reflect the structure of `Leffingwell`: the dataset contains 113 perceptual descriptors, approximately five times more than the other datasets, resulting in a substantially more fine-grained perceptual space. Such granularity may be difficult for LLMs to infer reliably through direct textual responses. Moreover, because the labels are binary, the task is closer to multi-label classification and may be less sensitive to subtle perceptual differences than continuous ratings. Together, these results suggest that embedding-based representations may be more suitable than direct generation for highly granular, binary-label odor prediction tasks.

Overall, these results suggest that LLMs encode a rich, linguistically grounded understanding of olfactory perception that generalizes across diverse datasets and rating schemes. They capture meaningful perceptual regularities that align with human judgments, highlighting the potential of language-based models as complementary tools to chemically grounded approaches. At the same time, the different performance across model families and datasets indicates that LLM-human alignment in olfactory prediction is likely shaped by architectural choices, post-training strategies, in-context adaptation, and task formulation, rather than by parameter scale alone.

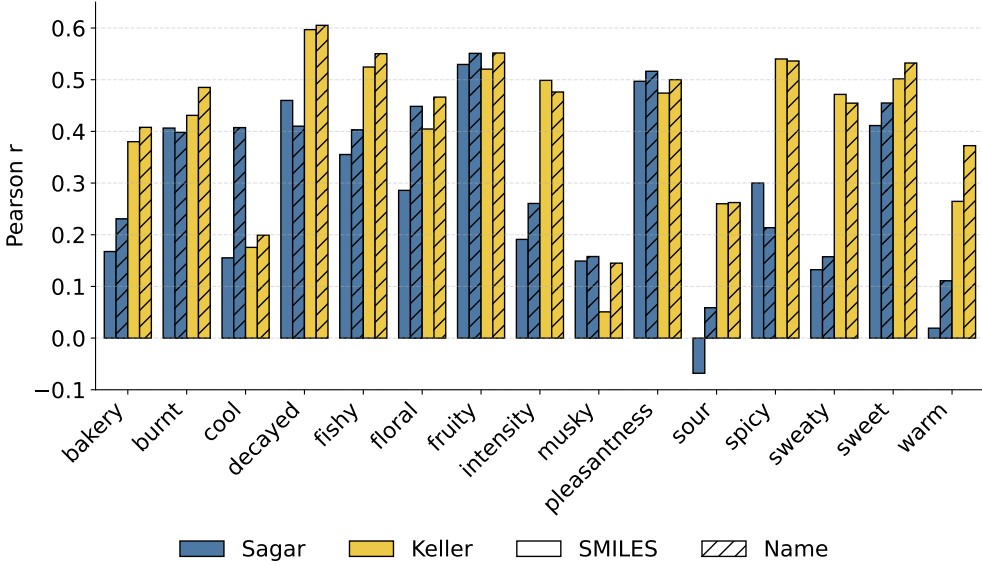

Figure 2: **Descriptor-level alignment patterns between human olfactory ratings and GPT-5-mini predictions across two input types**: molecular names and SMILES strings . Name-based inputs yield higher alignment for positive and concrete descriptors (e.g., fruity, floral, bakery, sweet, pleasantness, warm).

**Impact of Input Representation: Common Name vs SMILES.** Table 4 compares the performance of LLMs in predicting human olfactory responses based on two different input formats: common molecular names and SMILES strings. Across these models, prompts using odorant names consistently achieve higher or same alignment with human perceptual ratings than those using SMILES strings, for datasets with continuous or binary rating formats (i.e., `Leffingwell`, `Sagar`, and `Keller`). This pattern indicates that LLMs can effectively leverage prior and linguistic knowledge embedded in chemical or common names, which often implicitly encode information about chemical families (e.g., "aldehyde," "musk," "citrus") or sensory associations. Interestingly, this suggests that natural language alone can provide an amount of perceptually relevant information comparable to that extracted from explicit molecular structures by specialized chemical transformers. Whereas structure-based models learn such relationships through atom-level encodings, LLMs appear to infer them from the co-occurrence statistics of names and descriptive contexts in large text corpora. Thus, name-based prompts reveal an emergent form of semantic grounding in LLMs that bridges linguistic and perceptual knowledge domains.

For the `Snitz` dataset, which is inherently similarity-based, the pattern is reversed: SMILES-based inputs lead to higher alignment than name-based inputs. This difference may reflect the nature of the task. Unlike descriptor-rating datasets, where common names can provide useful semantic cues about individual odor qualities, `Snitz` requires estimating perceptual similarity between pairs of odorants. In this setting, explicit molecular structure may provide more useful relational information, since structural similarities and functional groups can help infer whether two molecules are likely to be perceived as similar. In contrast, common names may emphasize familiar or culturally learned odor associations for each molecule independently, but may provide weaker information about pairwise perceptual distances. This suggests that name-based prompts are particularly effective for predicting absolute odor qualities, whereas SMILES-based prompts may be more informative when the task requires comparing relationships between odorants

**Descriptor-Level Alignment Patterns Across Input Types.** We conducted an extra analysis to examine whether the alignment between LLMs and human judgments follows specific patterns at the level of individual perceptual descriptors. Figure 2 compares the alignment between human olfactory ratings and the predictions of GPT-5-mini across a set of shared perceptual descriptors between two datasets, using two input formats: common molecular names and SMILES strings. We specifically selected GPT-5-

mini for this analysis because, as shown in Table 4, the performance gap between these two input formats is particularly pronounced for this model. This comparison allows us to examine how linguistic versus structural representations influence the model's alignment with human olfactory perception.

The analysis reveals that for positive and concrete descriptors, those easily linked to familiar odor sources, such as fruity, floral, bakery, sweet, pleasantness, and warm, using the name-based input leads to a clear improvement in performance. In contrast, for negative or more abstract descriptors such as decayed, sour, sweaty, or burnt, the difference between the two input types is smaller and more consistent across input types. This asymmetry may reflect the fact that name-based representations carry richer semantic cues for positive and familiar smells, whereas negative or ambiguous odors are less frequently and less distinctly represented in everyday language.

**Assessing Potential Dataset Contamination.** Given that LLMs are pre-trained on large-scale web corpora, potential exposure to olfactory information cannot be fully excluded. However, not all such exposure constitutes data contamination. General knowledge about molecules, chemical families, odor descriptors, or compound–odor associations reflects the linguistic and scientific prior knowledge acquired during pretraining, and we do not consider this form of knowledge as problematic; rather, it is precisely the type of knowledge that may allow LLMs to reason about olfactory perception. By contrast, data contamination would involve direct or near-direct exposure to the evaluated benchmark records, such as the exact molecule–descriptor annotations or human rating values. This latter case is problematic because strong performance could reflect memorization of benchmark labels rather than genuine inference or generalization. We therefore assess this possibility using two complementary approaches.

First, we introduce a CID-only control experiment. The datasets used in our study include PubChem CIDs as separate metadata fields, and these identifiers are also widely accessible through public web resources. CIDs are database identifiers only; unlike SMILES strings, they do not intrinsically encode molecular structure, and unlike common names, they do not provide semantic cues about odor quality or chemical family. Therefore, strong alignment from CID-only inputs would raise the possibility of identifier-level recall or record-level memorization, rather than semantic or structure-based inference. Conversely, weak CID-only performance would argue against a simple memorization-based explanation for the observed results.

Table 5 summarizes the CID-only results across models. For the prediction-based datasets, namely `Keller`, `Sagar`, and `Leffingwell`, performance is random: correlations are near 0 for the continuous-rating datasets, and ROC-AUC values are close to 0.5 for the binary-label dataset. This suggests that the alignment observed in the main experiments is unlikely to be driven by simple identifier-based lookup or retrieval of records associated with PubChem identifiers. The main exception is `Snitz`, where CID-only performance is closer to the SMILES- and name-based results. Since `Snitz` is a pairwise similarity-based dataset, this result should be interpreted with caution. One possible explanation is that molecule-pair similarity records are more vulnerable to indirect memorization or retrieval than independent odorant-rating records. Another possibility is that, because CIDs are numerical identifiers, models may exploit superficial numerical patterns or comparisons between identifiers rather than relying on meaningful olfactory information. Overall, the CID-only results argue against identifier-based memorization for the prediction-based datasets, while the `Snitz` result remains less conclusive and suggests the need for additional controls in similarity-based settings.

Our second approach is to report results with the OLMo model throughout the paper. OLMo provides a useful reference point for assessing potential dataset contamination because its training data pipeline is substantially more transparent than that of many other model families. In particular, the publicly documented Dolma corpus applies filtering heuristics that remove files with `csv` and `json` extensions from the code-derived subset (Soldaini et al., 2024). This is relevant in our setting, as the olfactory datasets used in this study were released primarily as CSV files through public or access-controlled repositories. Although this does not rule out all possible exposure to molecule names, compound descriptions, or odor associations, it makes direct ingestion of the released benchmark files less likely. Thus, OLMo provides a useful, though not definitive, reference point for evaluating whether the observed alignment can be explained by direct dataset exposure.

Importantly, even if a model has not seen the benchmark files themselves, it may still have encountered individual compounds, common names, or qualitative odor associations through perfume websites, vendor

| Model | Keller | Sagar | Snitz | Leffingwell |
|---|---|---|---|---|
| Gemini-2.5-flash | $0.01 \pm 0.01$ | $0.05 \pm 0.03$ | $0.70 \pm 0.00$ | $0.50 \pm 0.00$ |
| GPT-5-mini | $-0.01 \pm 0.01$ | $-0.06 \pm 0.03$ | $0.73 \pm 0.00$ | $0.51 \pm 0.00$ |
| Gemma-4-31B-it | $0.02 \pm 0.01$ | $-0.01 \pm 0.02$ | $0.66 \pm 0.00$ | $0.50 \pm 0.00$ |
| Qwen-3.5-27B | $0.02 \pm 0.01$ | $-0.04 \pm 0.04$ | $0.74 \pm 0.00$ | $0.50 \pm 0.00$ |
| Llama-3.3-70B-Instruct | $0.01 \pm 0.01$ | $0.03 \pm 0.02$ | $0.69 \pm 0.00$ | $0.51 \pm 0.00$ |
| OLMo-3-1125-32B | $-0.02 \pm 0.01$ | $0.01 \pm 0.02$ | $0.66 \pm 0.00$ | $0.51 \pm 0.00$ |

Table 5: **Alignment scores across LLMs and datasets using CID-only input.** Results show that CID-only inputs yield near-random alignment for prediction-based datasets, suggesting limited evidence for memorization.

pages, fragrance forums, or chemistry resources. This possibility is especially relevant for name-based inputs, since recognizable compound names can activate prior lexical or world knowledge, whereas SMILES strings are much less likely to appear in natural descriptive odor-related text. However, such overlap is different from direct benchmark leakage. External sources typically provide coarse qualitative descriptions, such as whether a compound is floral, fruity, or sulfurous, rather than the structured quantitative labels used in our benchmarks. For example, several molecules may be described online as floral, but such sources usually do not specify how floral each molecule is on a shared numerical scale, nor do they provide the exact human ratings collected in controlled psychophysical datasets. In addition, the compounds in the perfume datasets are not generally the type of entries that are systematically documented with rich standardized descriptions on perfume websites, which more often focus on finished fragrances, fragrance notes, or a limited set of well-known aroma chemicals.

Taken together, the CID-only control and the OLMo comparison suggest that the main results are unlikely to be explained by simple memorization of released benchmark records. Instead, the stronger performance of name-based inputs is more plausibly explained by broader lexical priors and previously learned compound–odor associations. This interpretation is consistent with the empirical pattern that name-based inputs generally outperform SMILES-based inputs, while still distinguishing such prior knowledge from direct memorization of benchmark labels.

# 6 Discussion

The complexity of olfactory perception, from the perspectives of psychology, chemistry, neuroscience, and language, is undeniable. Despite decades of research, there is still no well-grounded, digitized framework for describing odorants, nor a fully developed linguistic system that captures the richness of olfactory experience. Humans, however, have found ways to communicate odors: sometimes by relating them to familiar sources (e.g., "rose-like," "burnt," "citrus"), and at other times through abstract perceptual dimensions that have emerged through language, such as pleasantness, intensity, or familiarity. In either case, the space of olfactory description remains deeply complex, and intertwined with the sciences of chemistry, cognition, and language.

In this work, we present a systematic analysis aimed at understanding the interaction between language, chemistry, and odor perception. To do so, we leveraged human perceptual datasets that describe how specific monomolecular odorants are experienced. We then prompted LLMs to provide similar perceptual judgments and compared their predictions with human data. Importantly, our analysis was not limited to LLMs trained exclusively on linguistic data, we also evaluated specialized molecular transformers trained directly on chemical structures, in order to examine whether the perceptual reasoning capabilities of LLMs extend beyond their linguistic training.

The results show that LLMs trained solely on natural language often outperformed chemically specialized transformers in predicting how humans perceive the smell of molecules. This finding suggests that a considerable amount of perceptual knowledge about odorants is implicitly encoded within human language.

Moreover, we observed systematic differences in performance depending on the form of input, when odorants were presented through chemical structures (e.g., SMILES) versus through their common names, revealing how linguistic and structural representations capture distinct but complementary aspects of olfactory meaning.

Together, these results point to a promising direction toward bridging the long-standing structure–odor gap: leveraging the rich semantic priors embedded in human language to model, interpret, and ultimately unify chemical and perceptual representations of smell.

**Limitations.** Our work is constrained by the limited availability of perceptual olfactory data. Only a few datasets currently provide human olfactory ratings, and these datasets cover a relatively narrow range of odorants. This limitation affects not only the scale of evaluation, but also the ability to assess generalizability to new odorants, out-of-distribution chemical spaces, and perceptual regimes not represented in existing benchmarks. More broadly, stronger controls for potential data contamination would ideally require newly collected perceptual datasets that are unlikely to have appeared in model pretraining corpora.

We also restricted our analysis to single, well-defined molecular odorants and excluded mixtures. Although mixture perception is an important direction, it remains a challenging open problem: including mixtures would make it difficult to determine whether model failures reflect weak olfactory alignment or limitations in how mixture components are represented and combined. This is particularly relevant because molecular models are trained primarily on single-molecule inputs. We therefore focus on mono-molecular odorants, where each stimulus has a clear chemical representation, enabling a fairer and more interpretable comparison between LLMs and molecular transformer models.

Another limitation concerns the comparison between large language models and molecular transformer models. Although chemical language models are specialized for molecular structures and are trained directly on molecular representations, they are currently much smaller than frontier large language models. More importantly, large-scale chemical foundation models comparable in size and capacity to modern LLMs have not yet been developed or made widely available. Therefore, differences in performance may reflect not only the type of input representation or training objective, but also differences in model scale, capacity, and maturity of the available model families. Future development of larger molecular foundation models would allow a more balanced comparison between general-purpose language models and structure-specialized molecular models

Finally, to assess the intrinsic capability of LLMs and ensure fairness in the information provided to both model types, we also intentionally avoided specialized prompting strategies. While more elaborate prompting or retrieval-augmented approaches may further improve LLM performance, our goal was to evaluate what olfactory information is accessible under a simple and comparable prompting setup. Future work could explore whether task-specific prompting, retrieval, or chain-of-thought-style reasoning improves olfactory prediction, particularly when paired with newly collected datasets and more diverse odorant classes.

**Future Work.** Future research should aim to address the current limitations of available olfactory datasets by generating larger perceptual datasets to include a wider range of odorants, containing both single molecules and complex mixtures. Incorporating mixtures is essential, as real-world olfactory experiences often emerge from interactions among multiple compounds that produce emergent perceptual qualities. Beyond data, an important direction lies in developing multimodal modeling frameworks that integrate chemical, linguistic, and behavioral information. Each modality provides a unique and complementary perspective: chemical representations capture the physicochemical basis of odor, linguistic descriptions encode human semantic and cultural understanding of smells, and behavioral responses reflect perceptual and emotional aspects of olfaction. Leveraging these modalities jointly could enable cross-domain supervision, where, for instance, linguistic embeddings guide molecular models toward human-relevant features, and chemical structure constrains language models to remain physically grounded. Such multimodal alignment holds promise for building unified models that not only predict human olfactory perception more accurately but also bridge the gap between symbolic, sensory, and chemical representations of smell.

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

| Dataset | Metric | GPT-5-mini | | Gemini-2.5-flash | |
|---|---|---|---|---|---|
| | | SMILES | Name | SMILES | Name |
| Keller | r | $0.36 \pm 0.03$ | **$0.40 \pm 0.03$** | $0.35 \pm 0.03$ | $0.40 \pm 0.03$ |
| Sagar | r | $0.25 \pm 0.04$ | **$0.32 \pm 0.04$** | $0.30 \pm 0.04$ | $0.36 \pm 0.03$ |
| Snitz | r | $0.70 \pm 0.00$ | **$0.73 \pm 0.00$** | $0.71 \pm 0.00$ | $0.70 \pm 1.71$ |

Table 6: **Alignment scores across LLMs and datasets for two different input types for the second template.** The difference between the two input formats is more pronounced for *Gemini*, whereas for *GPT*, the standard error decreases when using the name-based input format. Temperature for Gemini=1

## A  Prompt Templates and Ablations

To obtain perceptual ratings from LLMs, we designed structured text prompts that instruct the model to act as an olfactory rater.

Each prompt specifies the *molecule identity*, provided either as a common name or as a SMILES string, a list of *perceptual descriptors* to be rated (e.g., *sweet*, *woody*, *fishy*), and a predefined *rating range*. For similarity judgments, the prompt instead instructs the model to provide a numerical similarity score between two odorants.

The model is explicitly instructed to return only a valid JSON object containing numerical ratings for each descriptor, without any additional text or explanation. This structured output format ensures reliable parsing and consistent downstream analysis.

To evaluate the robustness of model responses to prompt framing, we conducted an ablation study using an alternative prompt template. In this variant, instead of instructing the model to *act as an olfactory rater*, the prompt asks the model to *imagine smelling the odorants*. This formulation more closely mirrors the instructions given to human subjects in psychophysical experiments.

Below, we provide both prompt templates for descriptor ratings and similarity judgments. The results obtained using the alternative template are reported in Table 6. All main results presented in the paper are based on the primary template.

### A.1  Prompt Templates

We designed two prompt templates to obtain perceptual judgments from LLMs. Each template was evaluated on two tasks: i) descriptor ratings and (ii) similarity ratings.

### A.1.1  Template 1: Olfactory Rater Instruction

**Case 1: Descriptor Ratings**

```
System message:
You are an olfactory rater. Output ONLY valid JSON.

User message:
Molecule:
- Name: vanillin

Descriptors (rate each from 0 to 100): ["sweet", "woody", "floral"]

Output rules:
- Return ONLY a single valid JSON object. No prose, no markdown.
- Keys must match the descriptor list exactly.
- Values must be numbers in [0,100].
```

**Case 2: Similarity Ratings**

```
System message:
You are an olfactory rater. Output ONLY valid JSON.

User message:
Molecule A:
- Name: vanillin

Molecule B:
- Name: ethyl vanillin

Task:
Rate the perceptual similarity between Molecule A and Molecule B.

Similarity scale:
- 0 = completely different odor
- 100 = identical odor

Output rules:
- Return ONLY a single valid JSON object. No prose, no markdown.
- Use exactly this key: "similarity"
- Value must be a number in [0,100].
```

### A.1.2   Template 2: Human Smelling Instruction (Ablation)

**Case 1: Descriptor Ratings**

```
System message:
Imagine you are smelling an odorant as a human participant in an olfactory experiment.
Output ONLY valid JSON.

User message:
Molecule:
- Name: vanillin

Descriptors (rate each from 0 to 100): ["sweet", "woody", "floral"]

Instructions:
Provide ratings as a human would based on perceived smell.

Output rules:
- Return ONLY a single valid JSON object. No prose, no markdown.
- Keys must match the descriptor list exactly.
- Values must be numbers in [0,100].
```

**Case 2: Similarity Ratings**

```
System message:
Imagine you are smelling two odorants as a human participant in an olfactory experiment.
Output ONLY valid JSON.

User message:
Molecule A:
- Name: vanillin
```

| Model | Keller | | Sagar | | Snitz | | Leffingwell | |
|---|---|---|---|---|---|---|---|---|
| | SMILES | Name | SMILES | Name | SMILES | Name | SMILES | Name |
| Qwen-3.5-4B | $0.03 \pm 0.02$ | $0.23 \pm 0.04$ | $0.06 \pm 0.03$ | $0.17 \pm 0.04$ | $0.74 \pm 0.00$ | $0.68 \pm 0.00$ | $0.52 \pm 0.01$ | $0.56 \pm 0.01$ |
| Qwen-3.5-27B | $0.17 \pm 0.04$ | $0.38 \pm 0.04$ | $0.21 \pm 0.05$ | $0.26 \pm 0.05$ | $0.64 \pm 0.00$ | $0.58 \pm 0.00$ | $0.61 \pm 0.01$ | $0.71 \pm 0.01$ |
| OLMo-3-7B-Instruct | $0.02 \pm 0.02$ | $0.05 \pm 0.03$ | $0.01 \pm 0.03$ | $0.05 \pm 0.04$ | $0.43 \pm 0.00$ | $0.65 \pm 0.00$ | $0.51 \pm 0.01$ | $0.55 \pm 0.01$ |
| OLMo-3-1125-32B | $0.04 \pm 0.02$ | $0.13 \pm 0.02$ | $0.06 \pm 0.02$ | $0.15 \pm 0.02$ | $0.52 \pm 0.00$ | $0.47 \pm 0.00$ | $0.64 \pm 0.01$ | $0.71 \pm 0.01$ |
| Gemma-4-E4B-it | $0.09 \pm 0.03$ | $0.30 \pm 0.03$ | $0.07 \pm 0.04$ | $0.19 \pm 0.03$ | $0.71 \pm 0.00$ | $0.72 \pm 0.00$ | $0.52 \pm 0.01$ | $0.60 \pm 0.01$ |
| Gemma-4-31B-it | $0.26 \pm 0.04$ | $0.40 \pm 0.03$ | $0.25 \pm 0.05$ | $0.33 \pm 0.00$ | $0.76 \pm 0.00$ | $0.62 \pm 0.00$ | $0.64 \pm 0.00$ | $0.71 \pm 0.01$ |

Table 7: **Alignment scores across datasets for LLMs at two model sizes within each model family.** For Keller, Sagar, and Snitz, the metric is Pearson correlation ($r$); for Leffingwell, the metric is ROC-AUC. For each model family, a smaller and a larger variant are compared to assess how model scale influences alignment scores.

```
Molecule B:
- Name: ethyl vanillin

Task:
Smell both odorants and rate their perceptual similarity as a human would.

Similarity scale:
- 0 = completely different odor
- 100 = identical odor

Output rules:
- Return ONLY a single valid JSON object. No prose, no markdown.
- Use exactly this key: "similarity"
- Value must be a number in [0,100].
```

Expected model output:

```
{
  "sweet": 90,
  "woody": 45,
  "floral": 72
}
```

## B   Effect of Model Size

In this study, we aimed to balance model scale with computational and cost affordability. For the open-source models, we therefore evaluated two variants from each model family: a smaller model and a larger model. This comparison allows us to examine how model size affects the alignment between model predictions and human olfactory judgments. Table 7 summarizes the results of this experiment. Overall, the results show that, within each model family, larger models tend to achieve higher alignment scores, suggesting that model scale contributes to improved olfactory alignment.

