# OpenReview forum: "Text2Smell: Emergent Representations of Human Olfactory Perception in Large Language Models"
_TMLR — Under review for TMLR_

### Review · Reviewer_A74Z · 2026-04-02

**Summary Of Contributions:**

This paper proposes the usage of pretrained large language models for categorizing smells of odorants based solely on their names or chemical (SMILES) description.

**Audience:**

Yes

**Audience Explanation:**

I do think the general concept of replacing/augmenting purpose built models for smell classification with LLMs may be interesting for some readers.

**Claims And Evidence:**

No

**Claims Explanation:**

The core concern with this paper is the leakage of information via the pretraining of the models: There is no way of guaranteeing that the datasets used for testing have not been observed in training already.
In fact, the results of OLMO on Snitz seem to imply this: Having the common name outperform the molecular structure description is a clear indicator for "cheating" by looking at its pretrained knowledge. The paper argues that this might be due to the models encoding information about chemical families, but that is much less likely than the model just knowing about the compound because it is present in its pretraining set. For example, Aldehydes can smell
- citrusy (e.g. hexanal, heptanal)
- almond-like (e.g. benzaldehyde)
- waxy/musky (e.g. long chain aldehydes)
- spicy/sweet (e.g. cinnamaldehyde, similar aromatic aldehydes)

There is nothing that fundamentally ties a common name to its smell, in fact there might be "false friends" that don't really smell like what their name would imply (methyl cinnamate doesn't really smell like cinnamon) since the chemical name stems from the structure not the smell.

There is certainly some overlap due to the connection

common name<-> chemical structure <-> smell

but the fact that we have a higher correlation with the common name (despite being one step removed from the smell defining element) implies that the model just amplifies its accumulated biases from the training data. The core question this opens up is whether these methods actually try to classify the odor, or just regurgitate folklore. Considering that this only captures a small set of monomolecules that come from  known benchmarking datasets, the likelihood for the latter is high.

In general, one should also look at simpler pretrained models (such as e.g. Word2Vec approaches) since those should also capture the correlations between odors and molecules. As an additional baseline one should also look beyond just linear probing: It is unlikely that the molecule<->odor connection is in a linearly seperable subspace. Training a small MLP shouldn't be too difficult and can already deal with some nonlinear extraction.

For the API models, one always has to report the version used (as in e.g. date of test) since the models behind the API are consistently updated without any warning or announcement (which is also a general reproducability concern). The set of LLMs tested also seem quite random: Why were exactly those models (all of which are considered weak in the grand scheme of things) chosen? Why not benchmark (open source) frontier models?

The labeling of Table 4 should also be adapted to note the exact model used: "Gemini-2.5-flash" is a very different model from the full size Gemini models. One should also report whether the models were allowed to "think" or not since that also affects performance.

As is, there is currently no way to judge whether these models actually have a concept of the smells (by e.g. looking at SMILES) or whether they just act as data aggregators that reproduce already known information (the latter would make them much less useful for researchers)

**Requested Changes:**

- Perform tests with nonlinear (e.g. MLP) heads
- Test more models, especially larger frontier models
- Study the degree of dataset contamination. This should be possible at the very least for Olmo, since the training set (Dolma 3) is open source
- Discuss the application to more complex smells (e.g. mixtures) since that reduces the odds of dataset contamination.

---

> ### Author Response · Authors · 2026-05-10
> **response to the review**
>
> We thank the reviewer for the feedback. We address the requested changes (C) as follow:
>
> **C1:** We intentionally used linear probes, following prior work, to evaluate the extent to which the learned representations encode olfactory perceptual information. We have clarified this choice in the “Alignment Evaluation for Transformers” section on page 7\.  The rationale for using linear probes is to avoid introducing additional model complexity, which would make it more difficult to determine whether the observed alignment arises from the representations of the original model or from the added complexity of the probing method. However, we agree that when the goal is to further improve alignment, non-linear probing methods woiuld be useful for capturing more complex relationships.
>
> **C2:** We have expanded our analysis by adding 3 additional open-source models and one additional closed-source model. We also included, in the Appendix, a comparison between smaller and larger models from the same family to analyze the effect of model size.
>
> **C3**: To address the concern of dataset contamination, we clarified the role of OLMo in our evaluation. Although OLMo was already included in the original experiments, we had not sufficiently explained why it is useful for assessing contamination. OLMo is trained on the Dolma dataset, which excludes CSV data files. Since the public olfactory benchmark datasets used in our study are hosted in public repositories and distributed as CSV files, OLMo provides a useful additional check against direct benchmark exposure. This makes it less likely that its performance is driven by having directly seen the released benchmark files during pretraining.
>
> As a second control, we introduced a CID-based experiment, where the input to the model is the PubChem chemical identifier rather than the molecule name or SMILES string. CIDs are publicly available identifiers and are also present in the original dataset repositories. Therefore, if the model were relying on memorized benchmark entries or simple identifier-based lookup, we would expect meaningful performance from CID inputs as well. However, the CID-based results were close to random, suggesting that the observed performance is unlikely to be explained by direct memorization of the benchmark data.
>
> We have also discussed existing olfactory data sources and distinguished between two types of possible data exposure: direct benchmark exposure and broader exposure to related olfactory information.
>
> To further discuss this concern in the paper, we have added a dedicated subsection on **“Assessing Potential Dataset Contamination”** to the Results section, where we discuss the possibility of benchmark exposure and present additional control analyses.
>
> **C4:** We agree that studying more complex odorants and odor mixtures is a very interesting direction. However, predicting perception from mixtures remains an even more challenging and open problem than the mono-molecular case. Including mixtures in the present study would therefore make it difficult to disentangle whether the observed results reflect the models’ olfactory alignment or limitations of the mixture-evaluation strategy itself. In particular, ambiguous results could arise because the approach used to represent or combine mixture components is inadequate, rather than because the models lack relevant olfactory information. This is especially important because molecular models are trained primarily on single-molecule inputs rather than odor mixtures.
>
> For this reason, we limited the scope of the present study to mono-molecular odorants, where the evaluation setting is better defined and more directly comparable across models. We have added this clarification to the limitations section and highlighted odor mixtures as a promising direction for future work.

---

> > ### Comment · Reviewer_A74Z · 2026-05-26
> >
> > Dear authors,
> >
> > I finally had time to properly analyze the changes you made and while I generally agree that the additional experiments do help, I still maintain that data contamination is a huge problem:
> > I underestand the reason for OLMo is due to it explicitly to prevent information leakage via CSV files, this does not prevent data contamination.
> >
> > Many of these compounds are well known, so even if the actual dataset is not 1:1 in the training set, this still means that the model relies on implicitly retrieving information from web-sources:
> > For example, you use "Phenethyl alcohol" as the example for "Molecular Representations" using the common name (page 4).
> > A simple web-search gives me the wikipedia https://en.wikipedia.org/wiki/Phenethyl_alcohol which notes:
> >
> > > Phenethyl alcohol is found in extract of rose, carnation, hyacinth, Aleppo pine, orange blossom, ylang-ylang, geranium, neroli, and champaca. It is also an autoantibiotic produced by the fungus Candida albicans.[6] Fusel alcohols like phenethyl alcohol are grain fermentation byproducts, and therefore trace amounts of phenethyl alcohol are present in many alcoholic beverages. It is therefore a common ingredient in flavors and perfumery, particularly when the odor of rose is desired.[3] It is used as an additive in cigarettes. It is also used as a preservative in soaps due to its stability in basic conditions. It is of interest due to its antimicrobial properties.
> >
> > So, even without leaking the dataset, the model can rely on memorization.
> > The inherent danger in the LLM approach is "are we actually classifying the odorants, or are we performing a web-search with extra steps".
> > The CID based approach doesn't really prove this since URLs (just like CSVs) are usually stripped during training (specifically during HTML linearization href targets are dropped).
> > However, I also do understand that there are limitations to what is practically doable: Having a completely fresh test set would be ideal, but this is not really tractable.
> >
> >
> >
> > The final limitation I see is the usage of linear probing heads: I agree that linear heads have the benefit of being very easy to train, but the same thing could be said of an e.g. 1 hidden layer MLP or even a LoRA.
> > The likelihood of a e.g. 6M parameter molecular transformer being able to project olfactory properties into a perfectly linear subspace seems to me very optimistic: How do we know that the results aren't the way they are because we just use a linear projection as the output layer?

---

> ### Author Response · Authors · 2026-05-27
> **response to the follow-up questions**
>
> Dear Reviewer,
>
> Thank you for taking the time to read our responses. Below, we address your follow-up questions regarding the experiments.
>
> Regarding the concern about memorization, we have clarified the distinction between two types of potential data exposure on page 10 of the manuscript (Assessing Potential Dataset Contamination). The type of exposure you refer to may not necessarily constitute benchmark data contamination. General web-based information can provide contextual knowledge that may help an LLM reason about an odorant, but it is fundamentally different from direct access to the human perceptual ratings used in our datasets. For example, the compound that you mentioned can be used as an additive in cigarettes without smelling like cigarettes, or may occur in alcoholic beverages without smelling like wine. More importantly, web sources do not generally provide the same descriptor-specific numerical ratings or pairwise similarity judgments collected from human participants in our benchmarks. Thus, even when contextual information about a compound exists online, it is not equivalent to exposure to the target labels evaluated in our experiments.
>
> Regarding the CID-based experiment, we suspect there may be a misunderstanding. PubChem Compound Identifiers (CIDs) are **explicitly included as separate columns in the benchmark data** files and are widely present across other web resources (and may or may not be included in URL). Our CID-based analysis was intended as a control for identifier-level memorization: because CIDs do not directly encode molecular structure or perceptual properties, strong performance from CID-only inputs would raise concerns about memorization of identifier-associated odor information or benchmark records.
>
> Finally, regarding linear probes, linear probing is a well-established methodology for assessing alignment between learned representations and target perceptual or neural measurements (described in the manuscript page 7 \- Alignment Evaluation for Transformer) \[1\]. Linear models have been widely used to evaluate representational alignment across **multiple modalities** \[1–5\]. Their use is motivated not by practical simplicity of training, but by their limited additional complexity. By avoiding the introduction of powerful nonlinear components that could independently learn the target mapping, linear models enable a more controlled assessment of how well the representations themselves align with the measured behavioral or neural responses. We would also like to emphasize that our results do not suggest that the target information is “perfectly” represented in a linear subspace. If this were the case, the observed correlations and ROC-AUC values would be expected to be 1\. Instead, the reported values indicate partial but meaningful linear accessibility of the relevant information.
>
> \[1\] Sucholutsky, I., Muttenthaler, L., Weller, A., Peng, A., Bobu, A., Kim, B., ... & Griffiths, T. L. Getting aligned on representational alignment. *Transactions on Machine Learning Research*.
>
> \[2\] Negi, A., OOTA, S. R., Nunez-Elizalde, A. O., Gupta, M., & Deniz, F. Brain-Informed Fine-Tuning for Improved Multilingual Understanding in Language Models. In *The Thirty-ninth Annual Conference on Neural Information Processing Systems*.
>
> \[3\] Moussa, O., Klakow, D., & Toneva, M. (2025, May). Improving semantic understanding in speech language models via brain-tuning. In *International conference on learning representations* (Vol. 2025, pp. 59823-59849).
>
> \[4\] Benara, V., Singh, C., Morris, J. X., Antonello, R. J., Stoica, I., Huth, A. G., & Gao, J. (2024). Crafting interpretable embeddings for language neuroscience by asking LLMs questions. *Advances in neural information processing systems*, *37*, 124137\.
>
> \[5\] Toneva, M., & Wehbe, L. (2019). Interpreting and improving natural-language processing (in machines) with natural language-processing (in the brain). *Advances in neural information processing systems*, *32*.

---

### Review · Reviewer_vieK · 2026-04-02

**Summary Of Contributions:**

In the proposed research authors investigate whether state-of-the-art LLMs (GPT, Gemini, OLMo) can predict human smell perception purely from linguistic cues and how their representations compare to those of molecular transformer models explicitly trained on chemical structure (MoLFormer-XL, MTL-BERT, ChemBERTa).

**Audience:**

Yes

**Audience Explanation:**

The core idea—probing LLMs with structured prompts and comparing to human ratings—is sound and timely.

**Broader Impact Concerns:**

No ethical concerns

**Claims And Evidence:**

No

**Claims Explanation:**

Clear task framing and dataset overview; the method diagram helps distinguish LLM prompting from molecular-transformer probing. However, there could be potential data contamination and memorization: public olfaction datasets and well-known molecule–descriptor associations likely appear in LLM pretraining corpora; no controls are provided to rule out memorization (e.g., name obfuscation, rare/novel molecules, paraphrase/alias controls).
Mismatch in evaluation regimes: GPT/Gemini are evaluated via generated ratings, whereas molecular encoders and OLMo are evaluated via fixed embeddings with linear probes.

**Requested Changes:**

The experimental design mixes generation-based evaluation for closed-source LLMs with representation probing for molecular encoders. This makes it difficult to attribute differences to model knowledge versus evaluation modality.

For name-based inputs, the hypothesis of linguistic priors is plausible, but without stronger contamination controls it remains equally plausible that LLMs are recalling known associations or near-duplicates (e.g., “civetone” → “musky”), especially given the ubiquity of perfumery resources online.

---

> ### Author Response · Authors · 2026-05-10
> **Response to the review**
>
> We thank the reviewer for the feedback. Below, we address the requested changes.
>
> **Dataset contamination.** To address the concern of dataset contamination, we clarified the role of OLMo in our evaluation. Although OLMo was already included in the original experiments, we had not sufficiently explained why it is useful for assessing contamination. OLMo is trained on the Dolma dataset, which, excludes CSV data files. Since the public olfactory benchmark datasets used in our study are hosted in public repositories and distributed as CSV files, OLMo provides a useful additional check against direct benchmark exposure. This makes it less likely that its performance is driven by having directly seen the released benchmark files during pretraining.
>
> As a second control, we introduced a CID-based experiment, where the input to the model is the PubChem chemical identifier rather than the molecule name or SMILES string. CIDs are publicly available identifiers and are also present in the original dataset repositories. Therefore, if the model were relying on memorized benchmark entries or simple identifier-based lookup, we would expect meaningful performance from CID inputs as well. However, the CID-based results were close to random, suggesting that the observed performance is unlikely to be explained by direct memorization of the benchmark data.
>
> To further discuss this concern in the paper, we have added a dedicated subsection on **“Assessing Potential Dataset Contamination”** to the Results section, where we discuss the possibility of benchmark exposure and present additional control analyses.
>
> **Generation-based vs. representation-based approaches.** We agree that the comparison between generation-based and representation-based approaches needs to be made clearer. Since embeddings cannot be extracted from closed-source models such as GPT and Gemini, and molecular foundation models are generally encoder-only and therefore not suitable for text generation, a fully matched comparison across all models is not possible. To make the evaluation clearer and more consistent, we changed the OLMo evaluation to a purely generation-based approach, so that all LLMs are evaluated using generation, while all molecular models are evaluated using embeddings. We also have clarified this in the manuscript.

---

### Review · Reviewer_6izo · 2026-05-01

**Summary Of Contributions:**

The paper asks whether LLMs can model human olfactory perception purely from text, and compares them against molecular transformer models trained on chemical structure. The key claim is that LLMs surprisingly align well (or better) with human perceptual judgments. The authors prompt LLMs (GPT, Gemini, OLMo) to generate perceptual ratings (e.g., pleasantness, sweetness) for odorants and compare these predictions against human psychophysical datasets. They further compare LLM performance with molecular transformer models trained on chemical structures (e.g., SMILES).

Strengths:
1. Well-motivated problem. It is a promising direction in the field of LLMs as models of human perception.
2. The core idea is clear and intriguing: Linguistic co-occurrence statistics may encode perceptual structure.
3. Empirical results support the effectiveness of llm for olfactory prediction.

Weakness:
1. This paper repeatedly suggest that LLMs “perceive the world as humans do”: I think the evidence in this paper only supports correlation, not real “Perception”. LLMs are predicting human ratings, not perceiving stimuli. Results may reflect language priors, not necessarily perceptual grounding.  When using molecule names, LLMs may rely on known associations (e.g., “vanillin” → sweet).
2. The generalization performance and out-of-distribution molecules are not investigated. So I suggest to provides evaluation on anonymized or synthetic molecule identifiers.
3. The comparison may not be fully fair. Molecular models are small (6M–45M parameters), and only use task-agnostic embeddings + linear probes. On the other hand, LLMs are massive and instruction-tuned with world knowledge.
4. Dataset coverage is limited, which is also acknowledged in limitations: small number of datasets; mostly monomolecular odorants; limited perceptual diversity.

**Audience:**

Yes

**Audience Explanation:**

see the review.

**Claims And Evidence:**

Yes

**Claims Explanation:**

see the review.

**Requested Changes:**

1. More studies on the generalization and ood performances.
2. More data coverage.
3. LLMs model human language about smells, not smells themselves. The paper should clearly disentangle perception vs description.

---

> ### Author Response · Authors · 2026-05-10
> **response to the review**
>
> We thank the reviewer for the feedback. Below, we address the identified weaknesses (W) and requested changes (C).
>
> **W1, C3:** We have revised the manuscript to address this point. Specifically, we changed the wording to emphasize “how language can be used to encode olfaction,” thereby avoiding claims that LLMs directly perceive odors and instead framing the results in terms of linguistic encoding and alignment with human olfactory judgments.
>
> **W2, W4, C1, C2:** Addressing these points thoroughly would require the collection of additional olfactory perception datasets. However, collecting such datasets with a sufficiently large number of stimuli is often a lengthy and demanding process due to sensory adaptation. Participants typically need to attend multiple sessions in order to provide perception ratings for many odorants with enough repetitions. In this study, we therefore selected four of the most representative and largest available datasets in this area, aiming to provide broad coverage despite existing data limitations. We have emphasized this limitation more clearly in the Limitation section and highlighted the importance of further data collection.
>
> **W3:** We have clarified this point in the limitation section. Although chemical language models are specialized for molecular structures, they are currently much smaller than large language models. We expect that the development of larger and more capable chemical models in the future may help overcome this limitation.

---

### Author Response · Authors · 2026-05-10
**General reponse to the reviews**

We thank the reviewers for the constructive feedback, which helped us strengthen our work and better position the scope of our claims. In response, we have changed and added the following:

* Added a dedicated data contamination subsection, including a clearer explanation of OLMo, a new CID-only control experiment, and a distinction between two types of possible data exposure, clarifying which one forms actual benchmark contamination.
* Added three additional open-source models, one additional closed-source model, and an appendix analysis of model-size effects.
* Clarified the generation-based versus representation-based evaluation setup across LLMs and molecular transformers.
* Revised the manuscript to reflect the new results, clarify ambiguous points raised by the reviewers, and discuss the limitations in more detail.